# Trends and Determinants of Antenatal Care Service Use in Ethiopia between 2000 and 2016

**DOI:** 10.3390/ijerph16050748

**Published:** 2019-03-01

**Authors:** Tensae Mekonnen, Tinashe Dune, Janette Perz, Felix Akpojene Ogbo

**Affiliations:** 1Translational Health Research Institute (THRI), School of Medicine, Western Sydney University, Penrith, NSW 2751, Australia; t.dune@westernsydney.edu.au (T.D.); j.perz@westernsydney.edu.au (J.P.); f.ogbo@westernsydney.edu.au (F.A.O.); 2School of Science and Health, Western Sydney University, Penrith, NSW 2751, Australia

**Keywords:** antenatal care service, trends, utilisation, Ethiopia, reproductive health, demographic and health survey

## Abstract

Antenatal care (ANC) services are an essential intervention for improving maternal and child health worldwide. In Ethiopia, however, ANC service use has been suboptimal, and examining the trends and factors associated with ANC service use is needed to inform targeted maternal health care interventions. This study aimed to investigate the trends and determinants of ANC service utilisation in Ethiopia for the period ranging from 2000 to 2016. This study draws on the Ethiopia Demographic and Health Survey data for the years 2000 (*n* = 7928), 2005 (*n* = 7276), 2011 (*n* = 7881) and 2016 (*n* = 7558) to estimate the trends in ANC service utilisation. Multivariate logistic regression models with adjustment for clustering and sampling weights were used to investigate the association between the study factors and ANC service utilisation. Over the sixteen-year period, the proportion of Ethiopian women who received the recommended four or more ANC visits increased from 10.0% (95% confidence interval (95% CI: 8.7–12.5%) in 2000 to 32.0% (95% CI: 29.4–34.3%) in 2016. Similarly, the proportion of women who received one to three ANC visits increased from 27.0% (95% CI: 23.6–30.7%) in 2000 to 62.0% in 2016 (95% CI: 60.4–67.3%). Multivariate analyses showed that higher maternal and paternal education, higher household wealth status, urban residency and previous use of a contraceptive were associated with ANC service use (1–3 and 4+ ANC visits). The study suggests that while Ethiopian pregnant women’s engagement with ANC services improved during the millennium development goal era (2000–2015), recommended ANC use remains suboptimal. Improving the utilisation of ANC services among pregnant women is essential in Ethiopia, and efforts should focus on vulnerable women.

## 1. Introduction

Globally, over 800 women die each day from complications related to pregnancy and childbirth [1], with Sub-Saharan Africa countries, including Ethiopia, accounting for more than 66 per cent of those deaths [1]. In 2015, the World Health Organization (WHO) estimated that 11,000 maternal deaths occurred during pregnancy, childbirth and the postpartum period in Ethiopia, or maternal mortality ratio of 353 (uncertainty interval: 247–575) [2], with substantial variations across regions of the country [3]. The main causes of maternal mortality in Ethiopia include postpartum haemorrhage, sepsis, pre-eclampsia and eclampsia and birthing complications [4]. Although most maternal deaths are preventable, many women often do not have access to evidence-based interventions such as the use of antenatal care (ANC) services during pregnancy, and related services in childbirth and the postpartum period due to poverty, lack of information, and cultural practices [5].

ANC is the service received by a woman during pregnancy from skilled health professionals [6]. Appropriate ANC visits have been shown to not only improve maternal and newborn health but also to increase maternal and newborn survival [7]. Consequently, WHO recommends that women who do not have any known pre-existing medical conditions should attend at least four ANC appointments [8]. The goal is to prepare women for birth and to prevent, detect, relieve, or manage health-related problems during pregnancy that may affect the mother and baby [9]. Despite the observed benefits of appropriate ANC utilisation globally [10], its use remains low in Ethiopia, with significant variations across regions [5,11,12,13]. Evidence from the 2016 Ethiopia demographic and health survey (EDHS) indicates that only a third of pregnant women receive four or more ANC visits, suggesting the need for studies to examine why Ethiopian women have a low prevalence of ANC use.

The recently concluded millennium development goal 5 (MDG) agenda sought to reduce maternal mortality by 75% between 2000 and 2015, and one of the strategies to achieve the goal was the improvement in ANC service use among women. While Ethiopia made efforts to reduce maternal mortality through improvement of ANC use [11], it remains unclear whether appropriate ANC use improved substantially in the context of the high maternal mortality in the country and the renewed global commitment to reduce maternal mortality to less than 70 per 100,000 births by 2030 (Sustainable Development Goal 3, SDG). Additionally, the assessment of factors associated with any change in the prevalence of ANC service use during the MDG period would be helpful to Ethiopian public health experts and policy-makers to refine current maternal health interventions for high-risk populations.

To the authors’ knowledge, no previous nationally representative studies have examined changes in the prevalence and factors associated with ANC service use during the MDG period to inform current priorities in Ethiopia. Therefore, the present study aimed to investigate the trends and determinants of ANC service use in Ethiopia between 2000 and 2016.

## 2. Methods

### 2.1. Data Sources

This study analysed data from the EDHS for the years 2000, 2005, 2011 and 2016, which were conducted by the Central Statistical Agency (CSA) under the guidance of the Ministry of Health (MOH) and technical assistance provided by the Inner City Fund (ICF) International USA. The EDHS was first conducted in Ethiopia in 2000 and a total of four national surveys have been administered to date. The survey provides relevant health and social information (including maternal and child health, trends in key health indicators in population) to guide policy decisions. A stratified two-stage cluster sampling procedure was used to recruit the nationally representative sample in all four surveys, with a high overall response rate that ranged from 95% to 98% [6,12]. Additional information about data collection, sampling and questionnaires used in the surveys are described in detail in the respective EDHS reports [6,12,13,14]. In the present study, a total of 30,643 (7928 in 2000, 7276 in 2005, 7881 in 2011 and 7558 in 2016) responses from reproductive age women were used.

### 2.2. Study Setting

The Federal Democratic Republic of Ethiopia has nine regional states, two city administrations, 611 *weredas* (districts) and 15,000 *Kebeles*. Regions are divided into zones, and zones, into administrative units called *weredas*. Each *wereda* is further subdivided into the lowest administrative unit called *kebele* (Population Census Commission, 2008). Ethiopia is the second most populous nation in Africa, with nearly 100 million people based on the 2007 census, after Nigeria with over 180 million people. Males represent 50.5% of the population and females represent 49.5%, with 21% in their reproductive ages (15–49 years). Although health service coverage reaches 92% of the population, the utilisation of maternal healthcare services is low. The 2016 EDHS report indicated that 62% of pregnant women received ANC care, 26% of pregnant women delivered in a health facility, and 17% received postnatal care. About 25% and 35% of all reproductive-age women and married women use contraceptives, respectively.

### 2.3. Outcome Variable

The frequency of ANC use was the outcome variable, defined according to the WHO recommendation. The use of ANC was measured as at least one visit to a doctor, nurse, midwife, or trained traditional birth attendant [10]. Women were grouped into three categories based on their utilisation ANC services: none, 1–3, and four or more visits.

### 2.4. Independent Variables

Anderson’s behavioural framework [15] was used to categorise the potential determinants of ANC utilisation. As described in Figure 1, the factors assumed to influence the use of ANC were categorised into four main factors: community level, predisposing, enabling and need factors. Community-level factors included residence (urban and rural), whilst predisposing factors included socio-demographic and health knowledge (mother’s age, household wealth index, mother’s and father’s education, mother’s marital status, mother’s employment status, frequency of reading magazine or newspaper, frequency of listening radio and frequency of watching TV). Enabling factors included permission to visit health services, distance to health facility, the presence of a companion and getting money to pay for health services, whilst factors such as contraceptive use and future plan to have a child were examples of need factors.

### 2.5. Analytical Strategy

Initial analysis involved the description of survey frequencies, followed by the analysis to investigate potential factors associated with the use of ANC for each year of the study. Multivariate logistic regression models were used to assess the association between the independent variables and the outcome variables for the years 2000, 2005, 2010 and 2016. A four-stage model was employed in the multivariable analyses, similar to the conceptual model described by Andersen RM [15] and used in previously published studies [16,17,18]. In the first stage, community-level factors were entered into the model to assess their relationship with the outcome variables. In the second model, the significant factors obtained from stage one model were added to socio-demographic and health service factors to examine their association with study outcomes. A similar approach was used for the enabling and need factors in the third and fourth stages, respectively. We also examined trends over the study period, stratified by each study variable in the models to assess the extent to which prevalence within study factors was increasing or decreasing. Any collinearity between study factors was also tested as appropriate but none was evident in the analysis.

Odds Ratios (and their corresponding 95% confidence intervals) were calculated as the measure of association between the independent variables and the outcome measures. All analysis were performed using the ‘svy’ command in Stata version 13.1 (Stata Corp, College Station, TX, USA) to allow for adjustments of sampling weights.

### 2.6. Ethics

This study used secondary data made publicly available by ICF International and the EDHS. The authorisation for using the data in the current study was granted from the DHS program upon presenting the aims of the study and the research plan. Detailed information on the data collection procedures employed by EHDS has been published as a full report elsewhere [6,12,13,14].

## 3. Results

### 3.1. Characteristics of the Study Participants

This analysis included 30,643 reproductive age women with a live birth 12 months preceding the survey. Between 2000 and 2016, the majority of participants had no formal education and were not employed (Table 1 and Table 2).

### 3.2. Trends in Antenatal Care Service Use in Ethiopia, 2000–2016

The proportion of reproductive age women who received recommended four or more ANC visits increased from 10.0% (95% confidence interval (95% CI): (8.74–12.5%) in 2000 to 32.0% in 2016 (95% CI: 29.4–34.3%). Similarly, women who received ANC increased from 27% (95% CI: 23.6–30.7%) in 2000 to 62.0% in 2016 (95% CI: 60.4–67.3%) (Figure 2).

### 3.3. Determinants of ANC Service Use: (1–3 Visits)

Between 2000 and 2016, women who had no education and whose partners had no schooling were less likely to have ANC visits (Table 3). The odds of ANC use among women from households of the highest wealth quintile were three times more likely to use ANC services compared to the lowest ones in all of the four surveys. Furthermore, women whose partners worked in agriculture (farming) were less likely to receive ANC services than those women with partners in non-agriculture jobs in 2000 and 2011 surveys. Women whose partners were not employed were also less likely to utilise ANC services compared to those women with partners in non-agriculture jobs (2011 and 2016). Those women who had used contraceptives were less likely to use ANC services compared to those who had not used contraceptives across all four surveys. Across all rounds of the survey, women in rural areas were less likely to have used ANC services than their urban counterparts (*p* < 0.0001). The women who reported that the distance to health facilities was a major barrier were also less likely to utilise ANC services compared to those women who reported that the distance to a health facility was not a hindrance.

### 3.4. Determinants of ANC Service Use (4 or More Visits)

Use of ANC services decreased with decreasing maternal level of education in all of the four surveys. Mothers who had no schooling and primary educational level were less likely to use ANC services compared to mothers who had secondary and higher level of education. Only in 2011 and 2016, mothers who were employed were more likely to use four and more ANC services compared to mothers with no employment. Mothers who never read, never watched television and never listened to the radio were less likely to have had four and more ANC visits compared to those mothers who had exposure to magazines or television and Radio at least once a week. Those women who used contraceptives and women who did not want to have more children in the future were more likely to attend four or more ANC visits than those women who had not used contraceptives and women who want to have more children in the future in all rounds of the EDHS. Women whose partner worked in agriculture (farming) were less likely to attend four or more ANC visits compared to women with partners in non-agriculture jobs. Women in the wealthiest group (high and middle) were found to be significantly likely to receive four or more ANC visits than those women in the poorest group between 2000 and 2016. Rural women were also less likely to have had four or more ANC care visits compared to those who lived in urban areas in all of the four surveys (Table 4).

## 4. Discussion

Findings from this study show that the proportion of women who received at least one ANC visit increased from 27% in 2000 to 62% in 2016. The proportion of reproductive age women who received four or more ANC visits also increased from 10% in 2000 to 32% in 2016. Rural residence, high and middle household wealth, contraceptive use, non-agricultural occupation, not wanting another baby and mass media engagement were factors significantly associated with attending at least 1-3 and four or more ANC visits.

Overall, although the use of ANC services increased substantially from 27% to 62% over the study period, the percentage of women who had four or more ANC visits did not increase significantly. These findings suggest the need for interventions to improve the inadequate utilisation of ANC service in Ethiopia. Increasing pregnant women’s utilisation of ANC service is indispensable as it also improves women’s engagement with skilled delivery, postnatal care and contraceptive services [19,20]. ANC visits also provide a window of opportunity to give appropriate information to women and their families about pregnancy care, safe childbirth, and postnatal recovery, including care of the newborn, promotion of early, exclusive breastfeeding, and assistance with family planning and contraceptive use [10]. Interventions may include service linkages, Behavioural Change Communication (BCC) and mobile health interventions [21]. For example, in Bangladesh focused BCC interventions given to mothers during ANC visits improved knowledge regarding danger signs of pregnancy and enjoyment with skilled care [22]. Education enables women to develop the confidence to make decisions about their own health and educated women also try to find quality health care that helps them get better care. In agreement with other studies done in South Ethiopia [23,24], women who did not have more than a primary level education were less likely to have ANC visits compared to those who completed secondary school or above across all four surveys. This suggests that addressing the women’s needs for schooling and decreasing school dropout rates may increase utilisation of ANC care.

The current study reveals that women who were living in rural areas were less likely to receive ANC services compared to those who resided in urban areas. This is in line with the results of other studies which reported that women from urban areas used ANC two times more than women from rural areas [25,26]. The lack of access to health care services in rural areas due to the inaccessibility of health facilities and professionals, lack of transportation services, limited finances, infrastructure and services may explain the difference [26]. As such, educational and health service efforts should be targeted at disadvantaged rural women [27]. Multivariate analysis indicated that women who had a history of contraceptive use before pregnancy were significantly likely to attend four or more ANC visits than those who did not. Findings from other studies also show a positive relationship between contraceptive use and ANC utilisation, which signifies the importance of linking contraceptive support and maternity care services [28].

Those mothers who did not have intention to become pregnant in the future were found to be less likely to receive ANC service than those who wanted children. This indicates the importance of making pregnancies planned through the provision of contraceptive services in reducing pregnant women’s disengagement from the health system [29]. Learning to use contraceptives requires increasing one’s knowledge about reproduction and their own bodies [30]. This education empowers women to have control and choice over their own bodies and futures and can translate into future engagement with the health care system for themselves and their children [31].

Our results show that women from wealthier households were six times more likely to use ANC compared with women from poorer households. This finding is consistent with studies from Nigeria and India which examined household wealth and ANC use [32,33]. This indicates a need to target low socio-economic status women with interventions to improve their utilisation of ANC care. For instance, providing them with some incentives and allocating more resources in rural areas may help improve their utilisation [26].

This study has some limitations that should be highlighted. Data in the primary study were subjected to recall bias as it refers to events five years past the survey date. As such, the proportions reported in this study may not reflect the actual value. In addition, DHS data on health services are limited to assessing availability and utilisation, with limited data collection about the actual quality of care women received. As such, associations between quality of care and ANC service utilisation could be assessed, which warrants further investigation using primary data collection approaches.

## 5. Conclusions

Overall, the findings of this study indicate that although the proportion of women who received ANC care increased over the sixteen-year period, the proportion of those who received the minimum recommended four visits remained low. Factors that significantly influenced the use of ANC in Ethiopia include urban residence, high and middle household wealth index, secondary and above level of education, history of contraceptive use before pregnancy and plans not to have any more children in the future. This suggests that interventions aimed at individual, community and health system levels are needed to improve pregnant women’s engagement with ANC care in Ethiopia, with a special focus on rural, poor and uneducated women. Finally, further research with the aim of assessing the quality of maternity care offered and examining the women’s and health care provider’s perceptions are needed. This would include exploration of barriers to the use of available services to ensure that the needs of pregnant women, their children and their families are met.

## Figures and Tables

**Figure 1 ijerph-16-00748-f001:**
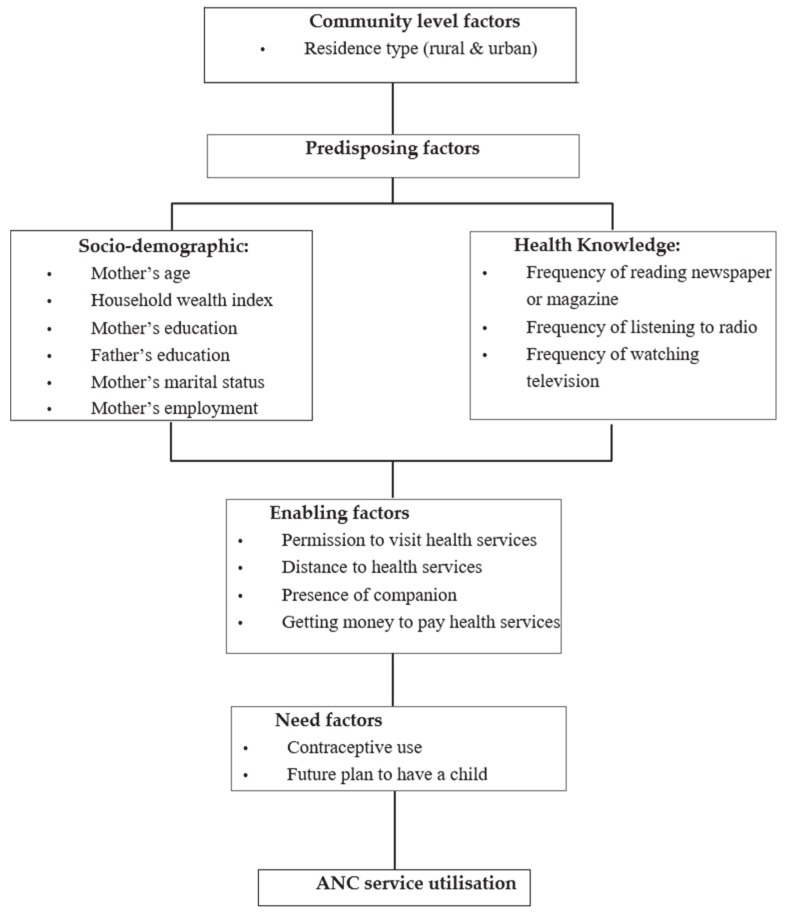
Conceptual framework adopted from Anderson’s (2003) model.

**Figure 2 ijerph-16-00748-f002:**
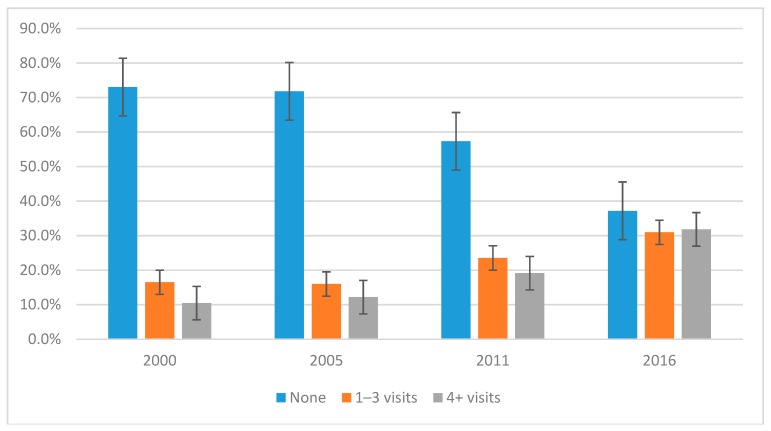
Trends in antenatal care service use in Ethiopia, 2000–2016.

**Table 1 ijerph-16-00748-t001:** Characteristics of the study participants who had one to three Antenatal care (ANC) visits in Ethiopia, 2000–2016.

Characteristics		2000 (*n* = 1308)	2005 (*n* = 1164)	2011 (*n* = 1856)	2016 (*n* = 2341)	*p* Trend
		*n* (%) *	*n* (%)	*n* (%)	*n* (%)	
**Community level factors**						
Residence type	Urban	201 (2.5)	93 (1.2)	352 (4.4)	262 (3.4)	
	Rural	1106 (13.9)	1071 (14.7)	1504 (19.0)	2079 (27.5)	*p* < 0.001
**Sociodemographic factors**						
Maternal age	15–24years	370 (4.6)	304 (4.2)	553 (7.0)	674 (8.9)	*p* < 0.001
	25–34	603 (7.6)	562 (7.7)	878 (11.1)	1159 (15.3)	
	35–49	334 (4.2)	297 (4.0)	425 (5.3)	507 (6.7)	
Marital status	Currently	1179 (14.8)	1081 (14.8)	1692 (21.4)	2226 (29.4)	
	Formerly	128 (1.6)	83 (1.1)	165 (2.0)	114 (1.5)	*p* < 0.001
Mother’s education	No education	985 (12.4)	825 (11.3)	1135 (14.4)	1412 (18.6)	
	Primary	233 (2.9)	274 (3.7)	626 (7.9)	745 (9.8)	*p* < 0.001
	Secondary and above	90 (1.1)	65 (0.8)	95(1.2)	183 (2.4)	*p* < 0.001
Household Wealth index	Poor	453 (5.7)	298 (4.0)	493 (6.2)	562 (7.4)	
	Middle	556 (7.0)	636 (8.7)	929 (11.7)	1412 (18.6)	*p* < 0.001
	Rich	298 (3.7)	229 (3.1)	435 (5.5)	366 (4.8)	*p* < 0.001
Mother’s employment	Not working	535 (6.7)	812 (11.7)	1287 (16.3)	1707 (22.5)	
	Working	769 (9.7)	301 (4.3)	569 (7.2)	634 (8.3)	*p* < 0.001
Partner education	No education	734 (9.4)	560 (7.7)	804 (10.3)	1013 (14.3)	
	Primary	377 (4.8)	414 (5.7)	820 (10.5)	906 (12.8)	*p* < 0.001
	Secondary and above	178 (2.2)	173 (2.3)	209 (2.6)	299 (4.2)	*p* < 0.001
Partner occupation	Agricultural	993 (12.5)	955 (13.1)	1372 (17.4)	1417 (18.7)	
	Non agricultural	300 (3.7)	196 (2.6)	461 (5.8)	664 (8.7)	*p* < 0.001
	Not working	14 (0.1)	12 (0.1)	23 (0.2)	260 (3.4)	*p* < 0.001
**Health knowledge factors**						
Frequency of reading newspaper or magazine	At least once a week	6 (0.07)	9(0.1)	41 (0.5)	42 (0.5)	
	Less than once a week	95 (1.2)	71 (0.9)	174 (2.2)	92 (1.2)	*p* < 0.001
	Never	1206 (15.2)	1080 (14.9)	1642 (20.8)	2207 (29)	*p* < 0.001
**Need factors**						
Contraceptive use	Yes	130 (1.6)	214 (2.9)	569 (7.2)	861 (11.3)	
	No	1178 (14.8)	949 (13.0)	1287 (16.3)	1480 (19.5)	*p* < 0.001
Intention to become pregnant	Immediately	751 (9.4)	693 (9.5)	1263 (16.0)	1757 (23.2)	
	Later	289 (3.6)	257 (3.5)	411 (5.2)	428 (5.6)	*p* < 0.001
	No more	268 (3.3)	211 (2.9)	182 (2.3)	155 (2.0)	*p* < 0.001

*n* *: weighted counts; the weighted total number and percentages vary between categories because of missing values.

**Table 2 ijerph-16-00748-t002:** Socio-demographic characteristics of the study participants who had four or more ANC visits.

Characteristics		2000 (*n* = 831)	2005 (*n* = 889)	2011 (*n* = 1508)	2016 (*n* = 2407)	*p* Trend
		*n* (%) *	*n* (%)	*n* (%)	*n* (%)	
**Community level factors**						
Residence type	Urban	396 (4.9)	346 (4.7)	541 (6.8)	603 (7.9)	*p* < 0.001
	Rural	435 (5.4)	543 (7.4)	967 (12.2)	1804 (23.8)	*p* < 0.001
**Socio-demographic factors**						
Maternal age	15–24	223 (2.8)	281 (3.8)	394 (4.9)	558 (7.3)	*p* < 0.001
	25–34	420 (5.3)	437 (6.0)	757 (9.6)	1321 (17.4)	
	35–49	187 (2.3)	170 (2.3)	357 (4.5)	528 (6.9)	
Marital status	Currently	734 (9.2)	800 (11)	1370 (17.3)	2234 (29.5)	
	Formerly	97 (1.2)	88 (1.2)	138 (1.7)	173 (2.2)	*p* < 0.001
Mother’s education	No education	403 (5.0)	453 (6.2)	644 (3.0)	1153 (15.2)	
	Primary	219 (2.7)	203 (2.7)	625 (7.9)	828 (10.9)	*p* < 0.001
	Secondary and above	209 (2.6)	232 (3.1)	239 (3.0)	426 (5.6)	*p* < 0.001
Household Wealth index	Poor	168 (2.1)	138 (1.8)	213 (2.6)	354 (4.6)	*p* < 0.001
	Middle	199 (2.5)	280 (3.8)	628 (7.9)	1313 (17.3)	*p* < 0.001
	Rich	464 (5.8)	471 (6.4)	667 (8.4)	740 (9.7)	*p* < 0.001
Mother’s employment	Not working	381 (4.8)	544 (7.8)	892 (11.3)	1646 (21.7)	
	Working	450 (5.6)	304 (4.3)	617 (7.8)	761 (10.0)	*p* < 0.001
Partner education	No education	279 (3.5)	276 (3.8)	428 (5.5)	770 (10.9)	
	Primary	222 (2.8)	258 (3.5)	699 (9.0)	908 (12.9)	*p* < 0.001
	Secondary and above	312 (3.9)	336 (4.6)	345 (4.4)	541 (7.6)	*p* < 0.001
Partner occupation	Agricultural	372 (4.6)	409 (5.6)	785 (9.9)	1186 (15.6)	*p* < 0.001
	Non agricultural	441 (5.5)	454 (6.2)	688 (8.7)	935 (12.3)	*p* < 0.001
	Not working	17 (0.2)	25 (0.3)	35 (0.4)	285 (3.7)	*p* < 0.001
**Health knowledge factors**						
Frequency of reading newspaper or magazine	At least	22 (0.2)	17 (0.2)	70 (0.8)	81 (1.0)	*p* < 0.001
	Less than	193 (2.4)	189 (2.6)	279 (3.5)	254 (3.3)	*p* < 0.001
	Never	599 (7.5)	668 (9.2)	1159 (14.7)	2071 (27.4)	*p* < 0.001
**Need factors**						
Contraceptive use	Yes	254 (4.9)	346 (4.75)	541 (6.8)	603 (7.9)	*p* < 0.001
	No	577 (5.4)	543 (7.4)	967 (12.2)	1804 (23.8)	*p* < 0.001
Intention to become pregnant	Then	465 (5.8)	534 (7.3)	1020 (12.9)	1845 (24.4)	*p* < 0.001
	Later	166 (2.1)	186 (2.5)	329 (4.1)	393 (5.2)	*p* < 0.001
	No more	198 (2.4)	169 (2.3)	159 (2.0)	168 (2.2)	*p* < 0.001

*n* *: weighted counts and the weighted total number varies between categories because of missing values.

**Table 3 ijerph-16-00748-t003:** Association between the study factors and ANC visits one to three in Ethiopia, 2000–2016.

Characteristics		2000 EDHS		2005 EDHS		2011 EDHS		2016 EDHS	
		*AOR (95% CI)	*p*-Value	AOR (95% CI)	*p*-Value	AOR (95% CI)	*p*-Value	AOR (95% CI)	*p*-Value
**Community level factors**									
Residence type	Urban	1.00		1.00		1.00		1.00	
	Rural	0.39 (0.24–0.61)	*p* < 0.001	0.65 (0.39–1.10)	*p* = 1.100	0.38 (0.26–0.56)	*p* < 0.001	0.42 (0.25–0.72)	*p* < 0.001
**Socio-demographic factors**									
Maternal employment status	Not working	1.00		1.00		1.00		1.00	1.00
	Working	1.00 (0.80–1.20)	*p* = 0.700	1.10 (0.80–1.40)	*p* = 0.500	1.00 (0.90–1.30)	*p* = 0.300	1.30 (1.10–1.60)	*p* = 0.020
Mother’s education	Secondary	1.00		1.00		1.00		1.00	
	Primary	0.80 (0.40–1.40)	*p* = 0.430	0.60 (0.38–1.16)	*p* = 0.150	0.83 (0.18–0.03)	*p* = 0.020	0.41 (0.21–0.82)	*p* = 0.010
	No education	0.40 (0.20–0.80)	*p* = 0.010	0.40 (0.24–0.73)	*p* = 0.002	0.28 (0.13–0.61)	*p* = 0.010	0.28 (0.14–0.57)	*p* = 0.010
Household wealth index	Poor	1.00		1.00		1.00		1.00	
	Medium	1.58 (1.27–1.96)	*p* = 0.010	1.60 (1.25–2.04)	*p* = 0.010	1.40 (1.13–1.73)	*p* = 0.010	1.38 (1.07–1.77)	*p* = 0.010
	Rich	3.21(2.34–4.40)	*p* = 0.010	2.23 (1.53–3.26)	*p* = 0.010	2.24 (2.24–4.54)	*p* = 0.010	2.30 (1.54–3.42)	*p* = 0.001
Partner occupation	Non-agricultural	1.00		1.00		1.00		1.00	
	Agricultural	0.62 (0.48–0.81)	*p* < 0.001	0.88 (0.65–1.21)	*p* = 0.580	0.62 (0.46–0.83)	*p* < 0.001	0.78 (0.59–1.02)	*p* = 0.700
	Not working	0.38 (0.14–1.08)	*p* = 0.070	0.77 (0.20–2.96)	*p* = 0.170	0.30 (0.13–0.69)	*p* = 0.010	0.62 (0.44–0.87)	*p* = 0.010
Partner education	Secondary	1.00		1.00		1.00		1.00	
	Primary	0.73 (0.51–1.04)	*p* = 0.089	0.58 (0.43–0.79)	*p* = 0.001	0.54 (0.36–0.81)	*p* = 0.003	0.61 (0.41–0.91)	*p* = 0.016
	No education	0.50 (0.35–0.70)	0.000	0.33 (0.24–0.46)	*p* < 0.001	0.34 (0.23–0.49)	*p* < 0.001	0.42 (0.29–0.61)	*p* < 0.001
Marital status	Currently married	1.00		1.00		1.00		1.00	
	Formerly married	0.98 (0.73–1.31)	*p* = 0.88	1.05 (0.75–1.48)	*p* = 0.77	0.86 (0.64–1.15)	*p* = 0.32	0.1 (0.05–0.19)	*p* < 0.001
**Need factors**									
History of contraceptive use	Yes	1.00		1.00		1.00		1.00	
	No	0.45 (0.32–0.62)	*p* < 0.001	0.64 (0.51–0.80)	*p* < 0.001	0.62 (0.50–0.77)	*p* < 0.001	0.72 (0.57–0.90)	*p* < 0.001
Wanted pregnancy at the time	Wanted then	1.00		1.00		1.00		1.00	
	Wanted later	1.15 (0.92–1.43)	*p* = 0.230	1.29 (0.99–1.68)	*p* = 0.060	0.98 (0.79–1.22)	*p* = 0.860	0.93 (0.70–1.22)	*p* = 0.580
	Unwanted	1.13 (0.87–1.47)	*p* = 0.340	0.97 (0.76–1.24)	*p* = 0.830	0.91 (0.67–1.24)	*p* = 0.550	0.56 (0.37–0.86)	*p* = 0.010
**Enabling factors**									
Seek permission to visit health services **	Not a big problem			1.00		1.00		1.00	
	A big problem			0.78 (0.62–0.97)	*p* < 0.001	0.97 (0.78–1.22)	*p* = 0.853	0.74 (0.57–0.97)	*p* = 0.029
Getting money to pay health services **	Not a big problem			1.00		1.00		1.00	
	A big problem			0.76 (0.61–0.96)	*p* = 0.021	0.91 (0.75–1.11)	*p* = 0.378	0.72 (0.56–0.91)	*p* = 0.008
Distance to health facility **	Not a big problem			1.00		1.00		1.00	
	A big problem			0.63 (0.50–0.80)	*p* < 0.001	0.81(0.66–1.01)	*p* = 0.066	0.62 (0.49–0.79)	*p* < 0.001
Accompany to health facility **	Not a big problem			1.00		1.00		1.00	
	A big problem			0.73 (0.59–0.91)	*p* = 0.007	1.17 (0.97–1.41)	*p* = 0.081	0.80 (0.65–1.00)	*p* = 0.052
**Health knowledge factors**									
Frequency of listening to Radio	At least once a week	1.00		1.00		1.00		1.00	
	Less than once a week	0.65 (0.39–1.10)	*p* = 0.110	0.68 (0.42–1.12)	*p* = 0.130	1.27 (1.00–1.63)	*p* = 0.050	1.15 (0.75–1.76)	*p* = 0.530
	Never	0.39 (0.23–0.68)	*p* < 0.001	0.51 (0.31–0.83)	*p* = 0.010	0.72 (0.55–0.95)	*p* = 0.020	0.61 (0.44–0.85)	*p* < 0.001

* Adjusted for community level, socio-demographic, media exposure, enabling and need factors. ** Variables not reported in the 2000 Ethiopia demographic and health survey (EDHS).

**Table 4 ijerph-16-00748-t004:** Association between the study factors and four or more ANC visits in Ethiopia, 2000–2016.

Characteristics		2000 EDHS		2005 EDHS		2011 EDHS		2016 EDHS	
		* AOR (95%CI)	*p*-Value	AOR (95% CI)	*p*-Value	AOR (95% CI)	*p*-Value	AOR (95% CI)	*p*-Value
**Sociodemographic level factors**									
Mother’s education	Secondary	1.00		1.00		1.00		1.00	
	Primary	0.73 (0.40–1.33)	*p* = 0.300	0.31 (0.17–0.54)	*p* < 0.001	0.23 (0.12–0.45)	*p* < 0.001	0.33 (0.17–0.62)	*p* < 0.001
	No education	0.34 (0.18–0.64)	*p* < 0.001	0.21 (0.12–0.38)	*p* < 0.001	0.13 (0.06–0.26)	*p* < 0.001	0.17 (0.09–0.34)	*p* < 0.001
House hold wealth status	Poor	1.00		1.00		1.00		1.00	
	Medium	1.38 (1.02–1.89)	*p* =0.040	1.41 (1.0–1.94)	*p* =0.030	2.01 (1.51–2.69)	*p* < 0.001	1.38 (1.07–1.77)	*p* =0.010
	Rich	6.12 (3.80–9.85)	*p* < 0.001	5.54 (3.75–8.20)	*p* < 0.001	6.04 (4.02–9.08)	*p* < 0.001	2.30 (1.54–3.42)	*p* < 0.001
Mother’s employment	Not working	1.00		1.00		1.00		1.00	
	Working	0.90 (0.66–1.22)	*p* = 0.500	1.54 (1.14–2.09)	*p* = 0.010	1.45 (1.16–1.80)	*p* < 0.001	1.30 (1.01–1.68)	*p* = 0.040
Partner occupation	Non-agricultural	1.00		1.00		1.00		1.00	
	Agricultural	0.32 (0.21–0.47)	*p* < 0.001	0.32 (0.23–0.44)	*p* < 0.001	0.40 (0.30–0.52)	*p* < 0.001	0.71 (0.51–0.98)	*p* = 0.040
	Not working	0.73 (0.23–2.30)	*p* = 0.590	1.52 (0.56–4.11)	*p* = 0.410	0.92 (0.27–3.12)	*p* = 0.890	0.48 (0.32–0.71)	*p* < 0.001
Partner education	Secondary	1.00		1.00		1.00		1.00	
	Primary	0.29(0.21–0.41)	*p* < 0.001	0.29(0.21–0.41)	*p* < 0.001	0.38(0.26–0.56)	*p* < 0.001	0.45(0.32–0.65)	*p* < 0.001
	No education	0.18(0.12–0.27)	*p* < 0.001	0.18 (0.12–0.27)	*p* < 0.001	0.19(0.13–0.28)	*p* < 0.001	0.26(0.18–0.36)	*p* < 0.001
Marital status	Currently married	1.00		1.00		1.00		1.00	
	Formerly married	1.21(0.80–1.83)	*p* = 0.35	1.33(0.91–1.93)	*p* = 0.14	0.79(0.54–1.15)	*p* = 0.22	0.1(0.05–0.19)	*p* < 0.001
**Need factors**									
Contraceptive use	Yes	1.00		1.00		1.00		1.00	
	No	0.15 (0.11–0.21)	*p* < 0.001	0.41 (0.30–0.56)	*p* < 0.001	0.35 (0.28–0.43)	*p* < 0.001	0.55 (0.42–0.70)	*p* < 0.001
Wanted pregnancy at the time	Wanted then	1.00		1.00		1.00		1.00	
	Wanted later	0.91 (0.68–1.21)	*p* = 0.520	1.14 (0.84–1.54)	*p* = 0.400	0.91 (0.69–1.19	*p* = 0.490	0.77 (0.59–1.02)	*p* = 0.070
	Unwanted	1.12 (0.82–1.54)	*p* = 0.470	1.04 (0.76–1.43)	*p* = 0.810	1.07 (0.76–1.50)	*p* = 0.710	0.60 (0.42–0.86)	*p* = 0.010
**Enabling factors**									
** Seek permission to visit health services	Not a big problem			1.00		1.00		1.00	
	A big problem			0.62 (0.48–0.80)	*p* < 0.001	0.62 (0.46–0.84)	*p* = 0.002	0.72 (0.53–0.97)	*p* = 0.031
** Getting money to pay health services	Not a big problem			1.00		1.00		1.00	
	A big problem			0.58 (0.44–0.76)	*p* < 0.001	0.73 (0.58–0.93)	*p* = 0.010	0.65 (0.52–0.83)	*p* = 0.010
** Distance to health facility	Not a big problem			1.00		1.00		1.00	
	A big problem			0.45 (0.35–0.59)	*p* < 0.001	0.49 (0.38–0.62)	*p* < 0.001	0.57 (0.43–0.75)	*p* < 0.001
** Accompany to health facility	Not a big problem			1.00		1.00		1.00	
	A big problem			0.60 (0.47–0.77)	*p* < 0.001	0.87 (0.69–1.10)	*p* = 0.272	0.62 (0.47– 0.81)	*p* = 0.001
**Community level factors**									
Residence type	Urban	1.00		1.00		1.00		1.00	
	Rural	0.18 (0.09–0.33)	*p* < 0.001	0.19 (0.12–0.30)	*p* < 0.001	0.30 (0.20–0.45)	*p* < 0.001	0.29 (0.15–0.55)	*p* < 0.001
**Health knowledge factors**									
Frequency of reading newspaper or magazine	At least once a week	1.00		1.00		1.00		1.00	
	Less than once a week	0.03 (0.02–1.25)	*p* = 0.080	0.80 (0.23–2.86)	*p* = 0.740	0.75 (0.24–2.33)	*p* = 0.620	0.89 (0.23–3.44)	*p* = 0.860
	Never	0.15 (0.00–0.20)	*p* < 0.001	0.23 (0.07–0.79)	*p* = 0.020	0.20 (0.07–0.57)	*p* < 0.001	0.31 (0.10–0.99)	*p* = 0.050
Frequency of listening to radio	Never	1.00		1.00		1.00		1.00	
	Less than once a week	0.55 (0.27–1.13)	*p* = 0.100	0.68 (0.42–1.12)	*p* = 0.130	1.15 (0.87–1.53)	*p* = 0.340	1.05 (0.69–1.61)	*p* = 0.810
	More than once a week	0.28 (0.14–0.59)	*p* < 0.001	0.51 (0.31–0.83),	*p* = 0.010	0.52 (0.39–0.69),	*p* < 0.001	0.51 (0.36–0.72)	*p* < 0.001
Frequency of watching television	At least once a week	1.00		1.00		1.00		1.00	
	Less than once a week	0.69 (0.18–2.72)	*p* = 0.600	0.21 (0.05–0.85)	*p* = 0.030	0.55 (0.39–0.78)	*p* < 0.001	0.57 (0.33–0.97)	*p* = 0.040
	Never	0.25 (0.06–0.94)	*p* = 0.040	0.07 (0.02–0.26)	*p* < 0.001	0.27 (0.18–0.40)	*p* < 0.001	0.35 (0.21–0.58)	*p* < 0.001

* Adjusted for community level, socio-demographic, media exposure, enabling and need factors. ** Variables not reported in the 2000 EDHS.

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
