# Peer review of "Trends and Determinants of Antenatal Care Service Use in Ethiopia between 2000 and 2016"

_ijerph, 2019, doi:10.3390/ijerph16050748_

Round 1

Reviewer 1 Report

Mekonnen et al.: Trends and determinants of antenatal care service use in Ethiopia

The article uses four rounds of Ethiopian Demographic and Health Survey to explore the trends in use and determinants of antental care services. The study shows that the coverage of antental care use increased substantially between 2000 and 2016, but still only one out of three women reaches the WHO recommendation of at least four visits. The data are of high quality, the analyses are sincerely done and the article has been well written. I do, however, have some comments, which should be taken into account before making the final decision of this paper.

1.      Abstract: Add year 2016 when referring to 32% (line 21).

2.      Introduction: Add also maternal mortality ratios so that the readers can compare the Ethiopian rates for example to the SDG goal of 70 per 100 000 live births, which is given later (lines 36-37).

3.      Introduction: Systematise organization (instead of organisation) for WHO (line 46).

4.      Methods: How did the authors handle multicollinarity and multiple comparisons?

5.      Analytical strategy: The reference to Andersen is incomplete (line 116).

6.      Results: The sentence relates to women who had used contraceptive compared to those who used contraceptives (lines 161-162).

7.      Discussion: Add % after “32” and “in 2016” (line 194).

8.      Table 1: Open “at least” and “less than” for example with a footnote. The reference to “then” seems to be related to intention to become pregnant, which is not that obvious in the current format of this table.

9.      Table 3: Correct marital status instead of marital statues.

10.   Some language correction is required. Some of the phrases could be revised, e.g. reproductive age women (line 135). I assume kebeles is the correct plural form of kebele (line 84). Double-check that you use plural forms for data (e.g. line 240).

Author Response

Reviewer reports:

Reviewer #1

The article uses four rounds of Ethiopian Demographic and Health Survey to explore the trends in use and determinants of antental care services. The study shows that the coverage of antental care use increased substantially between 2000 and 2016, but still only one out of three women reaches the WHO recommendation of at least four visits. The data are of high quality, the analyses are sincerely done and the article has been well written. I do, however, have some comments, which should be taken into account before making the final decision of this paper.

Abstract: Add year 2016 when referring to 32% (line 21).

Response: The year 2016 has been added to the abstract section. Please see page one.

Introduction: Add also maternal mortality ratios so that the readers can compare the Ethiopian rates for example to the SDG goal of 70 per 100 000 live births, which is given later (lines 36-37).

Response: The 2015 maternal mortality ration estimate for Ethiopia has been added:

In 2015, the World Health Organization (WHO) estimated that 11,000 maternal deaths occurred during pregnancy, childbirth and the postpartum period in Ethiopia or maternal mortality ratio of 353 (uncertainty interval: 247-575) [2], with substantial variations across regions of the country [3]. (Please see page one, line 36-39).

 Introduction: Systematise organization (instead of organisation) for WHO (line 46).

Response:  The text has been edited in the revised manuscript as requested by the reviewer (Page 1, line 36.)

Methods: How did the authors handle multicollinarity and multiple comparisons?

Response:

We thank the reviewer for observation. In the analysis, we tested for collinearity in the, consistent with past studies (Agho et al., 2017 & Ogbo et al., 2015). However, this was not noted in the original manuscript. In the analysis, we found no collinearity between study factors, hence, none was reported. For example, there was no cause to test and report collinearity for both maternal education and household wealth index as they were significant with the outcome in univariate analyses.

We have incorporated this text in the revised manuscript (Page 4, paragraph 1).

References:

1.       Agho KE, Ezeh OK, Ogbo FA, Enoma AI, Raynes-Greenow C. Factors associated with inadequate receipt of components and use of antenatal care services in Nigeria: a population-based study. Int Health. 2018;10(3):172-81. Epub 2018/03/22. doi: 10.1093/inthealth/ihy011. PubMed PMID: 29562242.

2. Ogbo FA, Agho KE, Page A. Determinants of suboptimal breastfeeding practices in Nigeria: evidence from the 2008 demographic and health survey. BMC public health. 2015;15:259. Epub 2015/04/08. doi: 10.1186/s12889-015-1595-7. PubMed PMID: 25849731; PubMed Central PMCID: PMCPMC4367831.

Please see page 4, line 125-126. 

Analytical strategy: The reference to Andersen is incomplete (line 116).

Response: The reference for Andersen has now been fixed, please see page four line 118.

Results: The sentence relates to women who had used contraceptive compared to those who used contraceptives (lines 161-162).

Response: The sentence has been revised to indicate the comparison groups.

Those women who had used contraceptive were less likely to use ANC services compared to those who had not used contraceptive across all four surveys.

See lines 162 and 163, page five.

Discussion: Add % after “32” and “in 2016” (line 194).

Response: The sign “%” has been added, please see page 13, line 194. 

Table 1: Open “at least” and “less than” for example with a footnote. The reference to “then” seems to be related to intention to become pregnant, which is not that obvious in the current format of this table.

Response: The variable health knowledge factors and responses has been modified to make sense. Please see table one.

Table 3: Correct marital status instead of marital statues.

Response: Again corrected, please see table one.

Some language correction is required. Some of the phrases could be revised, e.g. reproductive age women (line 135). I assume kebeles is the correct plural form of kebele (line 84). Double-check that you use plural forms for data (e.g. line 240).

Response: All errors identified by this author have also been fixed accordingly.

We would like to thank the reviewers for reading and providing feedback on our work.

Reviewer 2 Report

This very detailed study looks at factors related to the use of ANC in four epochs of time.  Limitations of the use of the first epoch and recall bias have been stated.  The study provides a variety of data that could be used to enhance the use of ANC

Minor comment: line 105 should be listening, not listing

Clarifications needed:

In the First Table, none of the variables add up to 100%; for example, rural vs urban.  Therefore, where do the others live?  These same discrepancies occur throughout.

Contraceptive use:  What does THEN mean?

How is the use of ANC decrease associated with the desire not to have another child abd therefore, no need for ANC just health care?

Author Response

Reviewer #2

This very detailed study looks at factors related to the use of ANC in four epochs of time.  Limitations of the use of the first epoch and recall bias have been stated.  The study provides a variety of data that could be used to enhance the use of ANC

Response:

We thank the reviewer for the comment.

Minor comment: line 105 should be listening, not listing

Response: Fixed, please see line 105 on page three.

Clarifications needed: In the First Table, none of the variables add up to 100%; for example, rural vs urban.  Therefore, where do the others live?  These same discrepancies occur throughout.

Response:  We thank the reviewer for the observation and note that it was an omission on our part. The proportions did up to 100% because of missing values as noted in the original manuscript (Table 1 footnote). We have clarified this point in footnote of Table 1 in the revised manuscript.

Contraceptive use:  What does THEN mean?

 Response: It is an editorial error commuted during formatting of Table 1. The response” THEN” belongs to the variable of pregnancy intention. It has not been replaced by “Immediately” to avoid confusion.

How is the use of ANC decrease associated with the desire not to have another child abd therefore, no need for ANC just health care?

 Response:  We found from the analysis that mothers who did not intend to become pregnant in the future were less likely to have used ANC services compared to those mothers who have planned to have additional children. In other words, the most recent pregnancy was unplanned and this might have discouraged them from accessing ANC and/or health care in general during pregnancy and postpartum periods. 

We would like to thank the reviewers for reading and providing feedback on our work.